# The Full-Genome Analysis and Generation of an Infectious cDNA Clone of a Genotype 6 Hepatitis E Virus Variant Obtained from a Japanese Wild Boar: In Vitro Cultivation in Human Cell Lines

**DOI:** 10.3390/v16060842

**Published:** 2024-05-24

**Authors:** Putu Prathiwi Primadharsini, Masaharu Takahashi, Tsutomu Nishizawa, Yukihiro Sato, Shigeo Nagashima, Kazumoto Murata, Hiroaki Okamoto

**Affiliations:** 1Division of Virology, Department of Infection and Immunity, Jichi Medical University School of Medicine, Shimotsuke, Tochigi 329-0498, Japan; thiwik8@jichi.ac.jp (P.P.P.); mtaka84@jichi.ac.jp (M.T.); tnishizawa@jichi.ac.jp (T.N.); shigeon@jichi.ac.jp (S.N.); kmurata@jichi.ac.jp (K.M.); 2Department of Internal Medicine, Kamiichi General Hospital, Nakaniikawa-Gun, Toyama 930-0391, Japan; sato@kamiichi-hosp.jp

**Keywords:** hepatitis E virus, HEV-6 variant, wild boar, cell culture, infectious cDNA clone, reverse genetics

## Abstract

Hepatitis E virus (HEV) can cause self-limiting acute and chronic hepatitis infections, particularly in immunocompromised individuals. In developing countries, HEV is mainly transmitted via drinking contaminated water, whereas zoonotic transmission dominates the route of infection in developed countries, including Japan. Pigs are an important reservoir for HEV infection. Wild boars, which share the same genus and species as domestic pigs, are also an HEV reservoir. During our nationwide study of HEV infection in wild boar populations in Japan, a genotype 6 (HEV-6) strain, wbJHG_23, was isolated in Hyogo Prefecture in 2023. The genomic length was 7244 nucleotides, excluding the poly(A) tract. The wbJHG_23 strain exhibited the highest nucleotide identity throughout its genome with two previously reported HEV-6 strains (80.3–80.9%). Conversely, it displayed lower similarity (73.3–78.1%) with the HEV-1–5, HEV-7, and HEV-8 strains, indicating that, although closely related, the wbJHG_23 strain differs significantly from the reported HEV-6 strains and might represent a novel subtype. The wbJHG_23 strain successfully infected the human-derived cancer cell lines, PLC/PRF/5 and A549 1-1H8 cells, suggesting that HEV-6 has the potential for zoonotic infection. An infectious cDNA clone was constructed using a reverse genetics system, and a cell culture system supporting the efficient propagation of the HEV-6 strain was established, providing important tools for further studies on this genotype. Using this cell culture system, we evaluated the sensitivity of the wbJHG_23 strain to ribavirin treatment. Its good response to this treatment suggested that it could be used to treat human infections caused by HEV-6.

## 1. Introduction

Hepatitis E virus (HEV) belongs to the family *Hepeviridae*, subfamily *Orthohepevirinae*, genus *Paslahepevirus* [1]. HEV is a single-stranded, positive-sense RNA virus that circulates in the blood as a membrane-cloaked quasi-enveloped virus but is shed in the feces as a naked non-enveloped virion [2]. The genome of HEV is approximately 7.2 kilobases (kb) and contains a short 5′-untranslated region (UTR) with a 7-methylguanosine cap, three major open reading frames (ORFs), and a short 3′-UTR terminated by the poly(A) tract [3,4]. ORF1 encodes a non-structural polyprotein containing multiple functional domains involved in viral replication [5,6,7]. ORF2 encodes the viral capsid protein, which can be produced in infectious, glycosylated, and cleaved forms; is essential during virion assembly and viral attachment to host cells; and is a major target for neutralizing antibodies [8,9,10,11]. ORF3 encodes a multifunctional phosphoprotein required for virion egress and has been reported to be a functional ion channel viroporin [12,13,14,15].

HEV infection is the leading cause of acute viral hepatitis worldwide. In addition to causing self-limiting acute hepatitis, HEV infection can also become chronic in immunocompromised patients [16]. In developing countries, HEV is mainly transmitted through the fecal–oral route via contaminated drinking water. Conversely, in industrialized countries, including Japan, the major transmission route of HEV infection is zoonotic food-borne via the consumption of undercooked or uncooked contaminated animal meat products or internal organs, such as the liver or intestines [17].

Members of the species *Paslahepevirus balayani* have been assigned to eight distinct genotypes: genotypes 1 HEV (HEV-1) to HEV-8 [1]. Certain members of the species *Paslahepevirus balayani* have been implicated in human infection. In addition, the species *Rocahepevirus ratti* has recently been reported to cause HEV infection in both immunocompromised and immunocompetent individuals [18,19,20].

Pigs serve as the primary reservoir of HEV. Most animal-derived HEV strains have been isolated from pigs. Besides pig-derived HEV strains, which can cross the species barrier and infect humans, HEV strains from other animals, including wild boars, deer, rabbits, camels, and rats, are also capable of causing zoonotic infections [21]. Wild boars have been associated with HEV-3 and HEV-4 infections. In addition, HEV-5 and HEV-6 have been isolated exclusively from wild boars in Japan [22,23,24,25]. Although the wild boar HEV-3 and HEV-4 can cause human infection [22] and HEV-5 has been reported to cause experimental zoonotic infection in non-human primates [26], the zoonotic potential of HEV-6 is unknown. Previously, two HEV-6 strains (wbJOY_06 and wbJNN_13) were identified in wild boars in Okayama and Nagano prefectures, respectively [23,24]. Unfortunately, due to the low viral load, these two HEV-6 strains were not multiplicated in cultured cells.

In the present study, we identified a third HEV-6 strain (wbJHG_23) with a high viral load from a wild boar in Hyogo Prefecture. The wbJHG_23 successfully infected human-derived cancer cell lines, suggesting that HEV-6 has the potential for zoonotic infection. An infectious cDNA clone was constructed using a reverse genetics system to provide a tool for the further study of this genotype. In addition, a cell culture system supporting the robust growth of HEV-6 was established in the present study. The wbJHG_23 strain responded well to ribavirin treatment in this cell culture system; therefore, ribavirin appears effective for treating individuals infected with HEV-6.

## 2. Materials and Methods

### 2.1. Collection of Serum and Liver Samples from Wild Boars

Serum and liver samples were procured from eight wild boars (*Sus scrofa leucomystax*) that had been captured between January and July 2023 in Hyogo Prefecture, situated in Kansai District in Japan (Figure 1). These samples were preserved at −80 °C until analyses.

A piece of liver tissue was washed three times with phosphate-buffered saline (pH 7.2) without Mg^2+^ and Ca^2+^ [PBS(-), Thermo Fisher Scientific, Inc., Waltham, MA, USA]. One hundred mg of the washed liver tissue was then finely minced using a razor blade and subsequently homogenized utilizing a BioMasher II (Nippi Incorporated, Tokyo, Japan) in the presence of 0.4 mL PBS(-). The resultant homogenate was then added with 0.6 mL of PBS(-) to make it into 10% (*w*/*v*) homogenate. The 10% homogenate was clarified via centrifugation in a high-speed refrigerated microcentrifuge (Tomy Seiko, Tokyo, Japan) at 21,130× *g* and 2 °C for 15 min, yielding a clear supernatant that was then stored at −80 °C until use.

### 2.2. The Enzyme-Linked Immunosorbent Assay (ELISA) for Detecting Anti-HEV Antibodies

To detect anti-HEV IgG antibodies in serum samples obtained from wild boars, an ELISA was performed. This assay utilized purified recombinant ORF2 protein from the HE-J1 strain (HEV-4) that had been expressed in silkworm pupae [27], following the method outlined in our earlier work [28]. A defined optical density (OD) value of 0.274, calculated as the mean plus six standard deviations, was established as the cut-off threshold for identifying swine anti-HEV IgG antibodies [28]. Samples exhibiting OD values equal to or greater than the cut-off were considered positive for anti-HEV IgG. The specificity of the anti-HEV assay was confirmed via absorption experiments involving identical recombinant ORF2 protein employed as the antigen probe, as detailed previously [28].

### 2.3. Qualitative and Quantitative Detection of HEV RNA

Reverse transcription (RT)–polymerase chain reaction (RT-PCR) was used to detect HEV RNA. Total RNA was extracted from either 100 µL of individual serum samples using a High Pure Viral RNA Kit (NIPPON Genetics Co., Ltd., Tokyo, Japan) and TRIzol-LS reagent (Thermo Fisher Scientific, Inc.) or from 50 mg of each liver specimen using TRIzol reagent (Thermo Fisher Scientific, Inc.), following the manufacturer’s instructions. The extracted RNA was reverse-transcribed with SuperScript IV reverse transcriptase (Thermo Fisher Scientific, Inc.), followed by nested PCR (ORF2/3-138 PCR) using TaKaRa Ex Taq (TaKaRa Bio, Shiga, Japan) and primers targeting the overlapping region of ORF2 and ORF3, as per a previously documented method [29]. Modified primers derived from well-conserved HEV sequences across all eight genotypes (HEV-1–8) [30] were used. The initial amplification product was 176 base pairs (bp), while the second-round amplification product was 138 bp in size.

For HEV genotyping, an additional RT-PCR assay employing primers targeting the ORF2 region (ORF2-457 PCR) [27] was conducted on ORF2/3-138 PCR-positive samples. The initial PCR amplification product measured 506 bp, and the second-round amplification product was 457 bp in size. The specificity of the RT-PCR assays was validated via a sequence analysis, as described below. The sensitivity of the RT-PCR assays was evaluated according to previously described procedures [29].

For the quantitation of HEV RNA, we conducted real-time RT-PCR using a previously described method [31], with a slight modification in the anti-sense primer that was specific for the HEV-6 strain identified in the present study (5′-AGG GGT TGG TTG GGT GGA-3′). To assess the reproducibility of the quantitative assay, each sample was tested in duplicate, and the mean value was utilized for subsequent analyses. The limit of detection of RT-PCR used in the current study was 2.0 × 10^1^ copies/mL.

### 2.4. Amplification of the Full-Length HEV Genome

In the present study, the complete sequence of the wbJHG_23 genome was determined. Total RNA extracted from 80 µL of 10% (*w*/*v*) liver homogenate from the wbJHG_23 boar was used for this purpose. Extraction was performed using the High Pure Viral RNA Kit, complemented by TRIzol-LS Reagent. Subsequently, cDNA synthesis was performed, followed by the nested PCR of four overlapping regions, including the extreme 5′- and 3′-terminal regions. Enzymes such as KOD -Multi & Epi- (Toyobo, Osaka, Japan) and TaKaRa LA Taq with GC Buffer (TaKaRa Bio) were utilized along with primers derived from highly conserved areas across all HEV-1–8 strains, for which the entire genomic sequences are known. Additional primers used during the amplification procedure were also incorporated. This methodology aligns with a previously documented approach [30]. The amplified regions, excluding the primer sequences, covered nucleotide (nt) positions 1–81 (81 nt), nt 41–3236 (3196 nt), nt 3205–5408 (2204 nt), and nt 5350–7244 (1895 nt) for the wbJHG_23 strain. The extreme 5′-end sequence (nt 1–81) was determined using a modified rapid amplification of cDNA ends (RACE) technique termed RNA ligase-mediated RACE (RLM-RACE) with the FirstChoice RLM-RACE kit (Thermo Fisher Scientific, Inc.), following previously established protocols [32]. The amplification of the 3′-end sequence [nt 5350–7244 (1895 nt), excluding the poly (A) tail] was conducted using the RACE method, as previously described [32].

### 2.5. The Determination and Analysis of Nucleotide Sequence

The amplification product was purified using the FastGene Gel/PCR Extraction Kit (NIPPON Genetics Co., Ltd.). Subsequently, both strands were subjected to direct sequencing or sequencing after cloning into the T-Vector pMD20 (TaKaRa Bio). Sequencing was performed using the Applied Biosystems 3130xl Genetic Analyzer (Thermo Fisher Scientific, Inc.) in conjunction with the BigDye Terminator v3.1 Cycle Sequencing Kit (Thermo Fisher Scientific, Inc.).

A sequence analysis was performed using the Genetyx software program (version 22; Genetyx Corp., Tokyo, Japan), while multiple alignments were generated using the MUSCLE (multiple sequence comparison by log-expectation) software program, version 3.5 [33]. Phylogenetic trees were constructed based on either the 412 nt ORF2 sequence (nt 5944–6355; accession no. M73218) or the entire sequence, employing the maximum-likelihood method with the Tamura–Nei model, facilitated by the MEGA11 software program (version 11.0.13) [34]. Cluster robustness was assessed by executing 1000 bootstrap replicates, and branches with bootstrap values exceeding 70% were grouped together [34]. For comparisons, the proposed reference sequences for HEV subtypes [35] were utilized.

### 2.6. Cell Culture

PLC/PRF/5 cells (ATCC No. CRL-8024; American Type Culture Collection, Manassas, VA, USA) and A549 1-1H8 cells (a subclone of A549, No. RCB0098; RIKEN BRC Cell Bank, Tsukuba, Japan) were cultured in growth medium consisting of Dulbecco’s modified Eagle’s medium (DMEM; Thermo Fisher Scientific, Inc.), supplemented with 10% heat-inactivated fetal bovine serum (FBS) (Thermo Fisher Scientific, Inc.), at 37 °C in a humidified 5% CO_2_ atmosphere, as previously outlined [36]. The growth medium for the cultivation of infected PLC/PRF/5 cells was supplemented with 1% dimethyl sulfoxide (DMSO) (Fujifilm Wako, Osaka, Japan). Following virus inoculation, the growth medium for infected A549 1-1H8 cells was substituted with maintenance medium comprising 50% DMEM and 50% medium 199 with Earle’s Salt (Thermo Fisher Scientific, Inc.), supplemented with 2% heat-inactivated FBS and 30 mM MgCl_2_.

### 2.7. Viruses

The liver homogenate supernatants from the wbJHG_23 boar (3.0 × 10^7^ copies/mL) were filtrated through 0.45 µm, followed by 0.22 µm, microfilters (Millex-GV; Merck-Millipore, Darmstadt, Germany) and utilized as inocula for propagation in PLC/PRF/5 and A549 1-1H8 cells (passage 0, P0). The culture supernatants of the inoculated A549 1-1H8 cells (9.2 × 10^7^ copies/mL) were filtered through a 0.22 µm microfilter and used as inocula for the subsequent propagation (P1) in PLC/PRF/5 and A549 1-1H8 cells. The culture supernatants containing HEV-6 progenies (1.7 × 10^8^ copies/mL) generated after the transfection of RNA transcripts of pwbJHG_23_P1 to PLC/PRF/5 cells were filtered through a 0.22 µm microfilter and used to inoculate PLC/PRF/5 and A549 1-1H8 cells.

### 2.8. Sucrose Density Gradient Centrifugation

The liver homogenate supernatants containing the wbJHG_23 strain (2.5 × 10^5^ copies) were subjected to equilibrium centrifugation in a sucrose density gradient, as described previously [37]. The gradients were fractionated, and the density of each fraction was measured using refractometry. Using a similar method, liver homogenate supernatants containing wbJHG_23 (5.0 × 10^5^ copies) were treated with 0.2% sodium deoxycholate (DOC-Na) and 0.2% trypsin in PBS(-) at 37 °C for 3 h according to the previously described method [2] and then subjected to equilibrium ultracentrifugation in a sucrose density gradient.

### 2.9. Virus Inoculation

Inoculation was carried out in a monolayer of PLC/PRF/5 or A549 1-1H8 cells in a six-well plate (Thermo Fisher Scientific, Inc.) with the indicated virus inoculum titers. The inoculum was diluted in PBS(-) containing 0.2% bovine serum albumin (BSA) at an inoculum volume of 200 µL/well. After incubation at room temperature for 1 h, the cells were washed five times with PBS(-), and 2 mL of the respective medium (as described in Section 2.6) was added to each well. The cells were then incubated at 35.5 °C in a humidified 5% CO_2_ atmosphere, as previously described [36]. Every other day, half of the culture medium (1 mL) was replaced with fresh medium. The collected culture supernatants were centrifuged at 1300× *g* at room temperature for 2 min, and the supernatants were stored at −80 °C until use.

### 2.10. Construction of a Full-Length Infectious cDNA Clone of wbJHG_23_P1

To generate a full-length infectious cDNA clone of wbJHG_23_P1, RNA was extracted from the culture supernatants containing wbJHG_23_P1 (derived from propagation in PLC/PRF/5 cells supplemented with 1% DMSO) using TRIzol-LS. Subsequently, cDNA synthesis was performed utilizing SuperScript IV along with the primer SSP-T (Table 1). Employing the synthesized cDNA as a template, two fragments spanning the entire wbJHG_23_P1 genome were amplified via PCR utilizing Platinum SuperFi II DNA Polymerase (Thermo Fisher Scientific, Inc.). Fragment 1 (F1) was amplified using the wbJHG_23_P1_1 and wbJHG_23_P1_2 primers, while fragment 2 (F2) was amplified using wbJHG_23_P1_3 and wbJHG_23_P1_4 primers (Table 1). Notably, the two overlapping amplified fragments were designed to possess a 15 nt homologous sequence. A cDNA clone of pJE03-1760F [38] served as the vector (pUC19 vector harboring the T7 promoter) for F1 and F2; this vector was linearized using inverse PCR, as per a previously described method [39], employing a high-fidelity DNA polymerase (KOD Plus ver. 2; Toyobo) and the wbJHG_23_P1_5 and wbJHG_23_P1_6 primers (Table 1). The resultant amplified fragments were purified using the FastGene Gel/PCR Extraction Kit.

For the construction of the full-length cDNA clone of pwbJHG_23_P1 under the T7 promoter, purified F1 and F2 were fused using the In-Fusion Snap Assembly method (TaKaRa Bio) following the manufacturer’s protocol. The plasmids were then extracted, and the sequences of the T7 promoter, full genome of pwbJHG_23_P1, and poly(A) tract were confirmed using the Sanger sequencing method.

In addition, as a negative control, pwbJHG_23_P1-GAA was synthesized through the mutation of the conserved RNA replication motif from GDD to GAA [Asp1568Ala (nt A4726C, nt A4727C) and Asp1569Ala (nt A4729C)]. The pwbJHG_23_P1 was used as the template. In brief, two fragments encompassing overlapping mutation sites within the wbJHG_23_P1 genome were amplified via PCR utilizing KOD Plus ver. 2. F1 GAA (nt 4054–4735, including the AvrII site to the mutation site) was amplified with primers wbJHG_23_P1_AvrII-F and wbJHG_23_P1-GAA-R, whereas F2 GAA (nt 4721–5744, including the mutation site to the StuI site) was amplified with the wbJHG_23_P1-GAA-F and wbJHG_23_P1_StuI-R primers. The vector was constructed by digesting the pwbJHG_23_P1 clone with AvrII and StuI (New England Biolabs, Ipswich, MA, USA) at nt 4065 and nt 5729, respectively. The digested cDNA and amplicons (F1 GAA and F2 GAA) were subsequently purified. To generate the full-length cDNA clone of pwbJHG_23_P1-GAA, the three fragments were fused using the In-Fusion Snap Assembly method. The sequence between the AvrII and StuI sites of pwbJHG_23_P1-GAA was confirmed through Sanger sequencing.

### 2.11. In Vitro Transcription and Transfection of RNA Transcripts to PLC/PRF/5 Cells

The full-length cDNA clone and its replication-defective mutant (pwbJHG_23_P1 and pwbJHG_23_P1-GAA, respectively) were linearized using BamHI-HF (New England Biolabs), followed by the synthesis of RNA transcripts with T7 RNA polymerase utilizing the AmpliScribeTM T7-FlashTM Transcription Kit (Biosearch Technologies, Hoddesdon, UK). Subsequently, the RNA transcripts of the cDNA clones underwent capping via the ScriptCapTM m7G Capping System (CELLSCRIPT, LLC, Madison, WI, USA). The integrity and yield of the synthesized RNAs were assessed using agarose gel electrophoresis. An aliquot (2.5 µg) of the capped RNA was transfected into confluent PLC/PRF/5 cells in a well of a 6-well plate employing the TransIT-mRNA transfection kit (Mirus Bio, Madison, WI, USA) according to the manufacturer’s instructions. After incubation at 37 °C for two days, the cells were washed with PBS(-), and the culture medium was substituted with 2 mL of growth medium containing 1% DMSO. The cells were then incubated at 35.5 °C. Every other day, half of the culture medium (1 mL) was replaced with fresh growth medium supplemented with 1% DMSO.

### 2.12. Western Blotting

To assess the expression of ORF2 and ORF3 proteins in PLC/PRF/5 cells following inoculation or transfection, protein samples from culture supernatants and cell lysates were subjected to sodium dodecyl sulfate–polyacrylamide gel electrophoresis (SDS-PAGE), then blotted onto polyvinylidene (PVDF) membranes (0.45 µm) (Merck-Millipore). Immunodetection was performed using either an anti-HEV ORF2 monoclonal antibody (MAb) (TA8117), generated against virus-like particles of HEV expressed in *Escherichia coli* (manuscript in preparation), or anti-HEV ORF3 MAb (TA0536) [37]. Detection was facilitated by HRP-conjugated Affinipure goat anti-mouse IgG (ProteinTech, Rosemont, IL, USA), and visualization and imaging were performed as described previously [38].

### 2.13. Immunofluorescence Assay

The PLC/PRF/5 cells, which had been inoculated or transfected, were seeded into four-well chamber slides (Watson, Tokyo, Japan) and underwent immunofluorescence staining following a previously described method [38] using anti-HEV ORF2 MAb (H6225) [31] and anti-HEV ORF3 MAb (TA0536) [37].

### 2.14. Sensitivity of HEV-6 to Ribavirin in Cultured Cells

The monolayers of PLC/PRF/5 cells were cultured in a 24-well plate (Thermo Fisher Scientific, Inc.). Each well was inoculated with 1.0 × 10^5^ copies/well of the cDNA-derived wbJHG_23_P1 in growth medium without FBS containing either 40 or 160 µM ribavirin (Fujifilm Wako) in DMSO (final concentration, 1%). The plate was then incubated at 37 °C for 2 h. Subsequently, the cells were washed five times with PBS(-), and 0.5 mL of growth medium containing 40 or 160 µM ribavirin in DMSO (final concentration, 1%) was added to each well, followed by further incubation at 35.5 °C. Every other day, half of the culture medium was replaced with fresh growth medium containing 40 or 160 µM of ribavirin in DMSO (final concentration, 1%). The concentrations of ribavirin utilized in the present study were selected based on our previous research assessing drugs’ effects on HEV growth in cultured cells, ensuring minimal cytotoxicity, and demonstrating the dose-dependent inhibition of HEV growth [40,41,42].

### 2.15. Nucleotide Sequence Accession Numbers

The nucleotide sequences of HEV isolates determined in the present study were deposited in the GenBank/EMBL/DDBJ databases with the following accession numbers: LC789535 (wbJHG_23), LC815003 (wbJHG_23_P1), and LC815004 (wbJHG_23_P1-GAA).

## 3. Results

### 3.1. Analyses of Anti-HEV IgG Antibodies and HEV RNA Prevalence in a Cohort of Eight Wild Boars

Serum and liver specimens from eight wild boars were tested to ascertain the presence of anti-HEV IgG antibodies and/or HEV RNA via ORF2/3-138 PCR. Of the examined specimens, only one boar (wbJHG_23) was seropositive for anti-HEV IgG, with an OD value of 0.442. The same boar demonstrated detectable HEV RNA in its serum, displaying a viral load of 4.1 × 10^3^ copies/mL. Furthermore, the boar identified as viremic also tested positive for HEV RNA in its liver specimen, with a viral load of 3.2 × 10^8^ copies/g.

### 3.2. Genetic Diversity of the Boar HEV Strain Isolated from an HEV-Infected Wild Boar

HEV RNA was also detected via ORF2-457 PCR in the serum sample of the wbJHG_23 boar. The amplification product of ORF2-457 PCR (412 nt, excluding primer sequences at both ends) was sequenced and subsequently compared with reference sequences proposed by Smith et al. [35]. A pair-wise comparison based on the 412 nt ORF2 sequence revealed that the wbJHG_23 strain shares its highest nucleotide sequence identity of 82.5–83.0% with HEV-6. Conversely, its similarity to the remaining seven genotypes ranged only between 76.7 and 81.8% (Table 2).

### 3.3. Characterization of wbJHG_23, an HEV-6 Strain 

The wbJHG_23 strain had a genomic length of 7244 nt, excluding the poly(A) tract at the 3′ terminus, and possessed three major ORFs, akin to HEV strains identified in humans and other animals [43]. Based on evidence that ORF2 and ORF3 proteins are translated from a single bicistronic subgenomic RNA [44], wbJHG_23 is presumed to encode an ORF1 protein of 1710 amino acids (aa) (nt 24–5153), an ORF2 protein of 660 aa (nt 5195–7174), and an ORF3 protein of 112 aa (nt 5187–5522). Interestingly, the N-terminus of the ORF3 protein is one or two aa shorter than those of HEV-1–4, HEV-7, and HEV-8 but aligns with two HEV-5 strains and two HEV-6 strains identified thus far.

The 5′-UTR of wbHG_23 was 23 nt, two nt shorter than the reported HEV strains, including HEV-6 strains (wbJOY_06 and wbJNN_13), due to the deletion of two nucleotides (GT) after nt 9 (Figure 2). Conversely, its 3′-UTR sequence consisted of 67 nt (nt 7178–7244) [excluding the poly(A) tail], a length akin to that of reported HEV-6 strains (wbJOY_06 and wbJNN_13), with the 3′-UTR sequence spanning 70–71 nt.

A comparative analysis with reference HEV-1–8 sequences, as proposed by Smith et al. [35], revealed that the wbJHG_23 strain shared the highest nucleotide sequence identity (80.3–80.9%) across the entire genome and 79.1–79.4%, 83.7–83.9%, and 89.6–90.6% within ORF1, ORF2, and ORF3, respectively, with two HEV-6 strains (wbJOY_06 and wbJNN_13) (Table 3). In contrast, it demonstrated lower similarity (73.3–78.1%) to other HEV-1–5, HEV-7, and HEV-8 strains, suggesting that, while closely related, the wbJHG_23 strain is divergent from the reported HEV-6 strains.

A phylogenetic analysis based on the entire genome (Figure 3) showed that the wbJHG_23 strain formed a cluster with the wbJOY_06 and wbJNN_13 strains, supported by a bootstrap value of 100%. Furthermore, separate phylogenetic trees based on the ORF1, ORF2, and ORF3 sequences (Figure 4) corroborated the clustering of wbJHG_23 with wbJOY_06 and wbJNN_13, supported by bootstrap values of 100%, 99%, and 95%, respectively, thereby categorizing wbJHG_23 within HEV-6.

### 3.4. The Ability of HEV-6 to Infect Human-Derived Cancer Cell Lines

To examine whether or not HEV-6 has the potential to cause human infection, a liver specimen of the wbJHG_23 boar that was positive for HEV-6 RNA was examined for the presence of virus particles. Liver homogenate supernatants containing wbJHG_23 strain were subjected to equilibrium centrifugation in a sucrose density gradient. The density peaked at 1.16 g/mL and 1.27 g/mL (Figure 5A), consistent with the density of membrane-associated HEV particles derived from culture supernatants [14,37] and membrane-unassociated HEV particles derived from feces [45,46], respectively. To confirm that both forms of HEV particles were present in the liver homogenate supernatants containing wbJHG_23, the supernatants were treated with 0.2% DOC-Na and 0.2% trypsin to remove the lipid membrane and ORF3 protein on the surface of the virus particles, respectively, at 37 °C for 3 h and then subjected to equilibrium centrifugation in a sucrose density gradient. Only a single peak density was observed at 1.26 g/mL (Figure 5B), which is consistent with that of the membrane-unassociated HEV particles [45,46], demonstrating that the density of 1.16 g/mL corresponds to the membrane-associated HEV particles. These results indicate that both membrane-associated and membrane-unassociated HEV particles were present together in the liver of the wbJHG_23 boar.

To examine whether or not HEV-6 can infect human-derived cancer cell lines, the liver homogenate supernatants containing wbJHG_23 were inoculated into PLC/PRF/5 derived from a hepatocellular carcinoma cell line (Figure 5C) and A549 1-1H8 derived from a lung adenocarcinoma cell line (Figure 5D), with the viral inoculum titers of 1.2 × 10^6^, 6.0 × 10^5^, and 1.2 × 10^5^ copies/well representing 5-, 10-, and 50-fold dilutions of the original titer in the liver homogenate supernatants, respectively. The HEV RNA titer in the culture supernatants of the wbJHG_23-inoculated PLC/PRF/5 cells increased dose-dependently and reached a peak titer of 1.1 × 10^8^ copies/mL at 56 days post-inoculation (dpi), 8.0 × 10^7^ copies/mL at 60 dpi, and 5.6 × 10^7^ copies/mL at 60 dpi for inoculum titers of 1.2 × 10^6^, 6.0 × 10^5^, and 1.2 × 10^5^ copies/well, respectively (Figure 5C). In addition, the liver homogenate supernatants containing wbJHG_23 successfully infected A549 1-1H8 cells (Figure 5D), where the HEV RNA titer in culture supernatants of the inoculated cells increased to reach a peak of 4.7 × 10^7^ copies/mL at 16 dpi, 5.3 × 10^7^ copies/mL at 16 dpi, and 5.0 × 10^7^ copies/mL at 20 dpi for inoculum titers of 1.2 × 10^6^, 6.0 × 10^5^, and 1.2 × 10^5^ copies/well, respectively.

Culture supernatants and lysates of PLC/PRF/5 cells from the last observation day (60 dpi) were subjected to Western blotting to examine the expression levels of HEV ORF2 and ORF3 proteins. Specific bands for ORF2 and ORF3 proteins were detected in the culture supernatants of wbJHG_23-infected cells and were undetectable in the culture supernatants of uninfected cells (Figure 5E, upper panels). Similarly, specific bands for ORF2 and ORF3 proteins were detected in the lysates of infected cells and undetectable in the lysates of uninfected cells (Figure 5E, lower panels). The observed variation in the lengths of ORF2 proteins in culture supernatants and cell lysates can be attributed to the presence and extent of the glycosylation of the viral protein [10,11]. In addition, the PLC/PRF/5 cells at 60 dpi were subjected to immunofluorescence assays to further examine the intracellular expression levels of HEV ORF2 and ORF3 proteins. The expression levels of ORF2 and ORF3 proteins were detected abundantly in wbJHG_23-infected cells, whereas the expression was undetectable in the uninfected control cells (Figure 5F). Collectively, these results demonstrate that HEV-6 can replicate efficiently in human-derived cancer cell lines and, therefore, has the potential to cause zoonotic infections in humans.

### 3.5. The Ability of HEV-6 Progenies to Produce Infectious Viruses in Cultured Cells

To examine the ability of the generated HEV-6 progenies to produce infectious virus, the culture supernatants of wbJHG_23-infected A549 1-1H8 cells were inoculated into PLC/PRF/5 (Figure 6A,B) and A549 1-1H8 (Figure 6C) cells with viral inoculum titers of 1.0 × 10^6^, 1.0 × 10^5^, and 1.0 × 10^4^ copies/well. For the propagation in PLC/PRF/5 cells, the growth medium was either not supplemented or supplemented with 1% DMSO (Figure 6A,B, respectively). The HEV RNA titer in culture supernatants of infected PLC/PRF/5 cells without DMSO supplementation increased gradually, peaking at 1.3 × 10^8^ copies/mL at 44 dpi, 1.3 × 10^8^ copies/mL at 48 dpi, and 2.5 × 10^7^ copies/mL at 48 dpi for inoculum titers of 1.0 × 10^6^, 1.0 × 10^5^, and 1.0 × 10^4^ copies/well, respectively (Figure 6A). In contrast, HEV replicated more efficiently for propagation in PLC/PRF/5 cells with 1% DMSO supplementation, in which the HEV RNA titer in culture supernatants reached its peak significantly earlier at 20 dpi (Figure 6B). The peak HEV RNA titers for inoculum titers of 1.0 × 10^6^, 1.0 × 10^5^, and 1.0 × 10^4^ copies/well were 1.3 × 10^8^, 1.5 × 10^8^, and 1.2 × 10^8^ copies/mL, respectively (Figure 6B), emphasizing the advantage of DMSO supplementation for propagation in PLC/PRF/5 cells. Similarly, the HEV RNA titer in culture supernatants of A549 1-1H8 cells inoculated with the HEV-6 progenies increased dose-dependently, peaking at 8.1 × 10^7^, 2.6 × 10^7^, and 1.3 × 10^7^ copies/mL at 12 dpi, 12 dpi, and 28 dpi for inoculum titers of 1.0 × 10^6^, 1.0 × 10^5^, and 1.0 × 10^4^ copies/well, respectively (Figure 6C).

The culture supernatants and lysates of PLC/PRF/5 cells (from propagation with DMSO supplementation) on the final day of observation (28 dpi) were subjected to Western blotting. Specific bands for ORF2 and ORF3 proteins were detected in the culture supernatants and lysates of the inoculated PLC/PRF/5 cells (Figure 6D). These results indicate that HEV-6 progenies generated in the culture supernatants of cells inoculated with liver homogenate supernatants containing wbJHG_23 can produce infectious viruses. In addition, the successful propagation of HEV-6 in PLC/PRF/5 and A549 1-1H8 cells indicated that the cell culture system used in the present study supported the efficient replication of HEV-6 in vitro—findings that will be useful for future studies concerning this genotype.

### 3.6. Construction of an Infectious cDNA Clone of HEV-6 and Transfection of Its RNA Transcripts to PLC/PRF/5 Cells

The availability of a highly replicating infectious cDNA clone of HEV-6 will facilitate various aspects of both fundamental and applied research concerning this genotype. A comparative analysis of the entire nucleotide sequences of the wild-type wbJHG_23 strain and its cell culture-produced variant (wbJHG_23_P1) revealed the presence of nine nucleotide differences (Table 4). Among these, two were non-synonymous (G1659A and C5461T), resulting in amino acid substitutions (G546S in the ORF1 region and T92I in the ORF3 region). The wbJHG_23_P1 sequence served as a template for constructing an infectious cDNA clone of HEV-6 (Figure 7A). Two fragments (F1 and F2) covering the whole genome of wbJHG_23_P1 were generated via RT-PCR and cloned into a pUC19 vector in a stepwise manner according to the In-Fusion cloning method (Figure 7B). A sequence analysis revealed that the resulting full-length cDNA clone was correctly constructed. To examine the ability of the resulting wbJHG_23_P1 cDNA clone to produce progeny viruses, the RNA transcripts of pwbJHG_23_P1 were transfected into PLC/PRF/5 cells. The HEV RNA titer in culture supernatants of the transfected cells gradually increased until 16 days post-transfection (dpt), peaking at 2.0 × 10^8^ copies/mL, and it was continuously maintained at a similar titer until the end of the observation period (28 dpt) (Figure 8A). In contrast, a gradual decrease in HEV RNA titer was observed in the culture supernatants of PLC/PRF/5 cells transfected with the RNA transcripts of a replication-defective mutant (pwbJHG_23_P1-GAA), which expressed functionally disrupted RNA-dependent RNA polymerase (RdRp) (Figure 8A).

Subsequently, the culture supernatants and lysates of transfected PLC/PRF/5 cells from 28 dpt were subjected to Western blotting. Specific bands of ORF2 and ORF3 proteins were detected in the culture supernatants and lysates of the cells transfected with RNA transcripts of pwbJHG_23_P1 and undetectable in the culture supernatants and lysates of pwbJHG_23_P1-GAA RNA-transfected cells (Figure 8B). Supporting this result, immunofluorescence imaging showed that the intracellular expression of ORF2 and ORF3 proteins was detected in the PLC/PRF/5 cells transfected with RNA transcripts of pwbJHG_23_P1 and undetected in the cells transfected with the RNA transcripts of pwbJHG_23_P1-GAA (Figure 8C). Taken together, these results demonstrate the replication ability of the HEV-6 cDNA clone.

### 3.7. The Infectivity of cDNA-Derived wbJHG_23_P1 Progenies to PLC/PRF/5 and A549 1-1H8 Cells

To examine whether or not the cDNA-derived wbJHG_23_P1 progenies generated in the culture supernatants were infectious, they were inoculated into PLC/PRF/5 (Figure 9A) and A549 1-1H8 (Figure 9B) cells at viral inoculum titers of 1.0 × 10^6^ and 1.0 × 10^5^ copies/well. The HEV RNA titer in culture supernatants increased gradually in both PLC/PRF/5 and A549 1-1H8 cells, peaking at 5.9 × 10^8^ copies/mL and 4.4 × 10^8^ copies/mL at 12 dpi in PLC/PRF/5 cells for inoculum titers of 1.0 × 10^6^ and 1.0 × 10^5^ copies/well, respectively (Figure 9A), as well as at 2.3 × 10^7^ copies/mL at 16 dpi and 1.4 × 10^7^ copies/mL at 20 dpi in A549 1-1H8 cells for inoculum titers of 1.0 × 10^6^ and 1.0 × 10^5^ copies/well, respectively (Figure 9B). Furthermore, specific bands for ORF2 and ORF3 proteins were detected in the culture supernatants and lysates of the inoculated PLC/PRF/5 cells from the final observation day (28 dpi) via Western blotting (Figure 9C). The expression of ORF2 and ORF3 (Figure 9D, upper panels) proteins was also detected in the infected cells, whereas the expression was undetectable in the uninfected control cells (Figure 9D, lower panels). These results indicated that the cDNA-derived wbJHG_23_P1 virus progenies were infectious.

### 3.8. Sensitivity of HEV-6 to Ribavirin Treatment in Cultured Cells

The HEV-6 has the potential to cross the species barrier and cause human infections. Ribavirin is currently used to treat certain cases of clinical HEV infection, such as chronic or acute fulminant hepatitis [47]. To examine the sensitivity of HEV-6 to ribavirin treatment, cDNA-derived wbJHG_23_P1 progenies were inoculated into PLC/PRF/5 cells in the presence of 40 or 160 µM ribavirin in DMSO (final concentration, 1%). The virus growth kinetics were observed for 28 days. The HEV RNA titer in culture supernatants of untreated control cells increased gradually and reached its peak at 3.4 × 10^8^ copies/mL by 24 dpi (Figure 10A). Treatment with ribavirin inhibited HEV growth in cultured cells in a dose-dependent manner. The HEV RNA titer in culture supernatants of the 160 µM ribavirin-treated cells decreased gradually and finally reached undetectable levels from 24 dpi onward (Figure 10A). Supporting this result, at 28 dpi, the intracellular expression levels of ORF2 and ORF3 proteins were abundant in the untreated control cells, (Figure 10B, upper panels), whereas they were undetectable in the cells treated with 160 µM ribavirin (Figure 10B, lower panels). These results indicate that HEV-6 is sensitive to ribavirin treatment in cultured cells and that ribavirin can be used to treat HEV-6-infected patients.

## 4. Discussion

The identification of novel HEV strains across an expanding range of animal species poses a burgeoning risk of zoonotic transmission, escalating public health concerns. In industrialized countries, including Japan, HEV infections predominantly originate from zoonotic sources [21]. Among these, wild boars, belonging to the same genus and species as the domestic pig—recognized as the primary reservoir of HEV—stand out as an important reservoir for HEV infection in humans. Initially, HEV infection in wild boars was exclusively attributed to HEV-3 and HEV-4 strains, not only in Japan [48,49,50,51,52] but also across various European and Asian countries [53,54,55], where they have been reported to cause zoonotic infections in humans. Subsequently, two novel genotypes, HEV-5 and HEV-6, were isolated exclusively from wild boar populations in Japan. Of note, experimental evidence has revealed that cDNA-derived progenies of HEV-5 successfully infected non-human primates (cynomolgus monkeys) [26], underscoring its potential for human infection; however, the zoonotic capacity of HEV-6 remains unclear.

In the present study, a wild boar (wbJHG_23) captured in Hyogo Prefecture (Figure 1) in 2023 was found to be positive for anti-HEV IgG, with detectable RNA present in both serum and liver samples. The wbJHG_23 strain exhibited the highest nucleotide sequence identity with two HEV-6 strains previously isolated from other prefectures in Japan: Okayama (wbJOY_06) [23] and Nagano (wbJNN_13) [24] (Table 2 and Table 3). However, unlike other HEV strains, the wbJHG_23 strain lacked two bases at nt 10 and 11 (Figure 2). Phylogenetic analysis based on the entire genome (Figure 3), as well as analyses based on ORF1, ORF2, or ORF3 (Figure 4), demonstrated that the wbJHG_23 strain formed a cluster with the two previously reported HEV-6 strains, further substantiating its classification within HEV-6. However, owing to the lower similarity demonstrated by wbJHG_23 to other HEV strains belonging to the remaining genotypes (HEV-1–5, HEV-7, and HEV-8), this newly identified strain is divergent from the two previously reported HEV-6 strains.

Considering the ability of wild-boar-derived HEV strains to cause infection in humans, HEV-6 might also have the potential for zoonotic transmission to humans. To test this hypothesis, we initially verified the presence of HEV particles in liver homogenate supernatants obtained from the wbJHG_23 boar (Figure 5A,B). The wbJHG_23 strain successfully infected two human-derived cancer cell lines, PLC/PRF/5 cells and A549 1-1H8, which are known for their high susceptibility to HEV infection. HEV RNA titers in the culture supernatants peaked at over 10^8^ copies/mL (Figure 5C,D), with both extracellular and intracellular expression of ORF2 and ORF3 proteins being detectable (Figure 5E,F). The ability of HEV-6 to infect human-derived cancer cell lines underscores its potential to cause human infections, similar to observations in other wild-boar-derived HEV genotypes [26,48,49,50,51,52,53,54,55]. Furthermore, virus progenies generated in the culture supernatants of infected cells retained their infectivity and successfully propagated to PLC/PRF/5 and A549 1-1H8 cells (Figure 6).

To establish a robust cell culture system, we introduced DMSO into the growth medium to propagate PLC/PRF/5 cells. Previous studies have noted the potential of DMSO to induce differentiation in tumor cells [56,57] and maintain differentiation in cultured primary liver cells [58]. A recent report underscored the essential role of DMSO supplementation in mitigating fate deterioration and facilitating adaptation to an in vitro environment [59]. In the context of propagating duck hepatitis B virus (DHBV), the addition of DMSO to the tissue culture medium has been shown to maintain high viral gene expression [60]. Furthermore, in the presence of DMSO, uninfected hepatocytes exhibited prolonged susceptibility to DHBV infection compared with untreated cell cultures [60]. In line with the previous findings, the current study successfully established a robust cell culture system by supplementing the growth medium with DMSO for the propagation of PLC/PRF/5 cells. This was evidenced by significantly enhanced virus replication (Figure 6B) compared to propagation in the same cell line without DMSO supplementation (Figure 6A). Notably, in cultures supplemented with DMSO, the time taken to reach the peak HEV RNA titer in the culture supernatants was shorter, even for the lowest inoculum titer of 1 × 10^4^ copies/well (Figure 6B, 20 dpi), than in cultures without DMSO supplementation (Figure 6A, 44 dpi).

Given the unavailability of the infectious cDNA clone of HEV-6, we generated one using the cell-culture-produced wbJHG_23_P1 variant as a template (Figure 7). The variant produced in cell culture was found to harbor nine nucleotide mutations in comparison to the wild-type wbJHG_23 (Table 4), suggesting the potential emergence of these mutations during cell culture. The HEV RNA titer in the culture supernatants of PLC/PRF/5 cells transfected with pwbJHG_23_P1 RNA transcripts peaked at over 10^8^ copies/mL and remained consistently high throughout the observation period (Figure 8A). A Western blotting analysis and immunofluorescence assays confirmed the intracellular and extracellular expression of ORF2 and ORF3 proteins (Figure 8B,C), validating the replication ability of the wbJHG_23_P1 cDNA clone. Furthermore, the progenies in culture supernatants derived from wbJHG_23_P1 cDNA clone were infectious (Figure 9). The availability of both the HEV-6 cDNA clone and the robust cell culture system established herein offers essential tools for various studies concerning this genotype, particularly investigations of factors related to cross-species transmission, replication kinetics, and drug resistance.

The 5′- and 3′-UTRs of RNA viruses have been reported to play pivotal roles in protein translation initiation and viral RNA replication [61,62,63,64]. The 5′-UTR of the HEV-1–6 and HEV-8 reference sequences was 25 nt long (Figure 2). Unlike other HEV strains, wbJHG_23 exhibited a deletion of two bases at the 5′-UTR at nt 10–11 (Figure 2). However, through the creation of an infectious cDNA clone, it was established that the wbJHG_23 HEV-6 strain retained its infectivity. The impact of this deletion in the 5′-UTR on HEV replication and infectivity would be an interesting topic for future investigation.

With the potential for zoonotic transmission carried by HEV-6, there is a continuous concern regarding the suitability of ribavirin, the currently used anti-HEV drug, for treating individuals infected with HEV-6. Using our robust cell culture system and the cDNA-derived wbJHG_23 progenies, we demonstrated that HEV-6 is sensitive to ribavirin treatment. Specifically, the HEV RNA titer in the culture supernatants of infected PLC/PRF/5 cells treated with 160 µM ribavirin gradually decreased and eventually reached an undetectable level (Figure 10A). This finding was further corroborated by immunofluorescence imaging, in which the expression of ORF2 and ORF3 proteins was undetectable (Figure 10B). Given the sensitivity of HEV-6 to ribavirin treatment in cultured cells, its potential application in the future antiviral treatment of patients infected with HEV-6 is plausible.

In conclusion, we identified a novel strain within HEV-6, designated as wbJHG_23, isolated from a wild boar in Hyogo Prefecture, Japan. The present study marks the first demonstration of the ability of HEV-6 to replicate in human-derived cancer cell lines, underscoring its potential to cause human infections. In addition, we successfully developed an infectious cDNA clone for HEV-6, along with a robust cell culture system, serving as a pivotal tool for various studies on this genotype, particularly those associated with the potential for cross-species transmission. The confirmation of the zoonotic transmission of HEV-6 necessitates further investigation through infection experiments involving non-human primates.

## Figures and Tables

**Figure 1 viruses-16-00842-f001:**
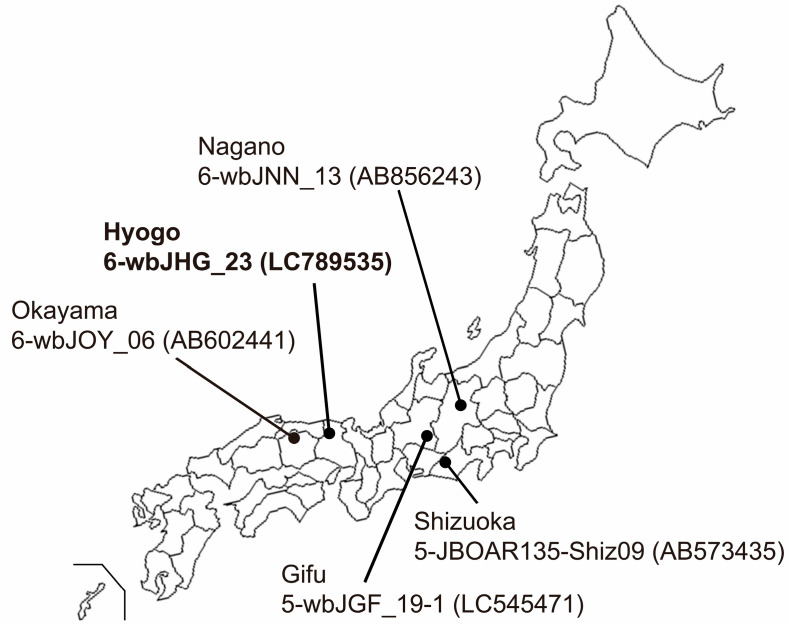
A map of Japan illustrating the precise positioning of Hyogo Prefecture, the site of origin for the novel HEV-6 strain (wbJHG_23 strain), which is highlighted in boldface for visual clarity. In addition, the map delineates four other prefectures where four reported HEV-5 and HEV-6 strains have been acquired from wild boars; each HEV strain is denoted by its genotype, isolate name, and accession number enclosed in parentheses.

**Figure 2 viruses-16-00842-f002:**
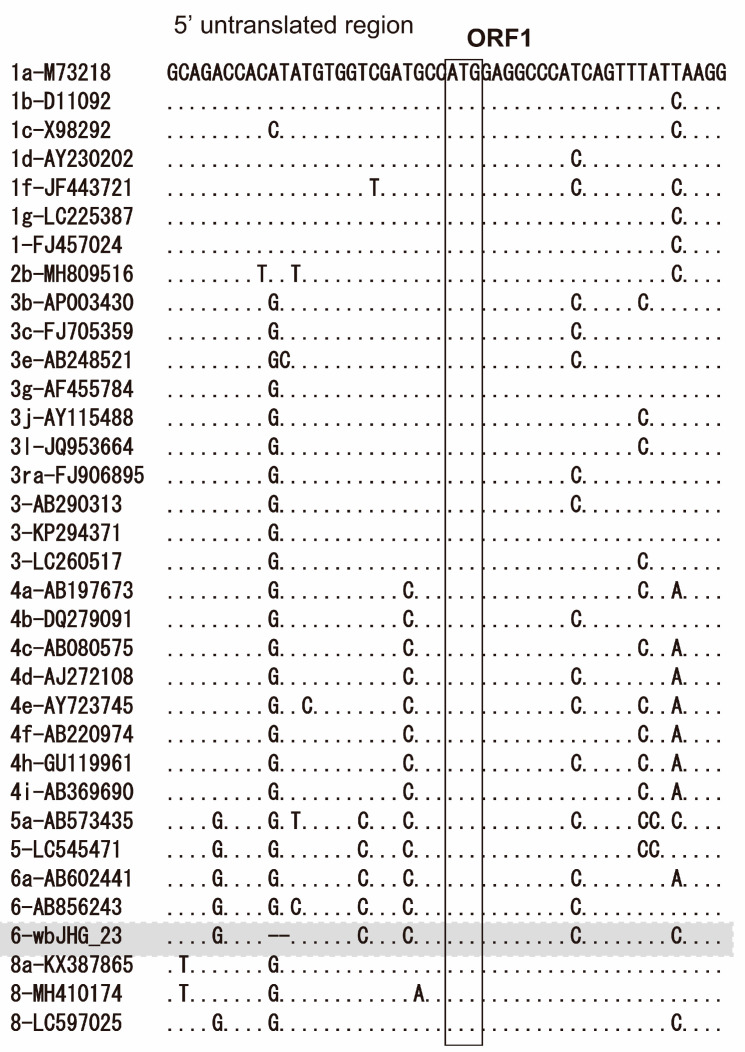
Alignment of the 5′-terminal 50-nucleotide sequences of 34 representative HEV strains for which the 5′-untranslated region (UTR) sequence is available. Each HEV strain is denoted with its genotype/subtype, followed by the accession number. The sequence for the HEV-6 strain identified in the present study is highlighted in gray. The initiation codon for open reading frame 1 (ORF1) is denoted within an open square.

**Figure 3 viruses-16-00842-f003:**
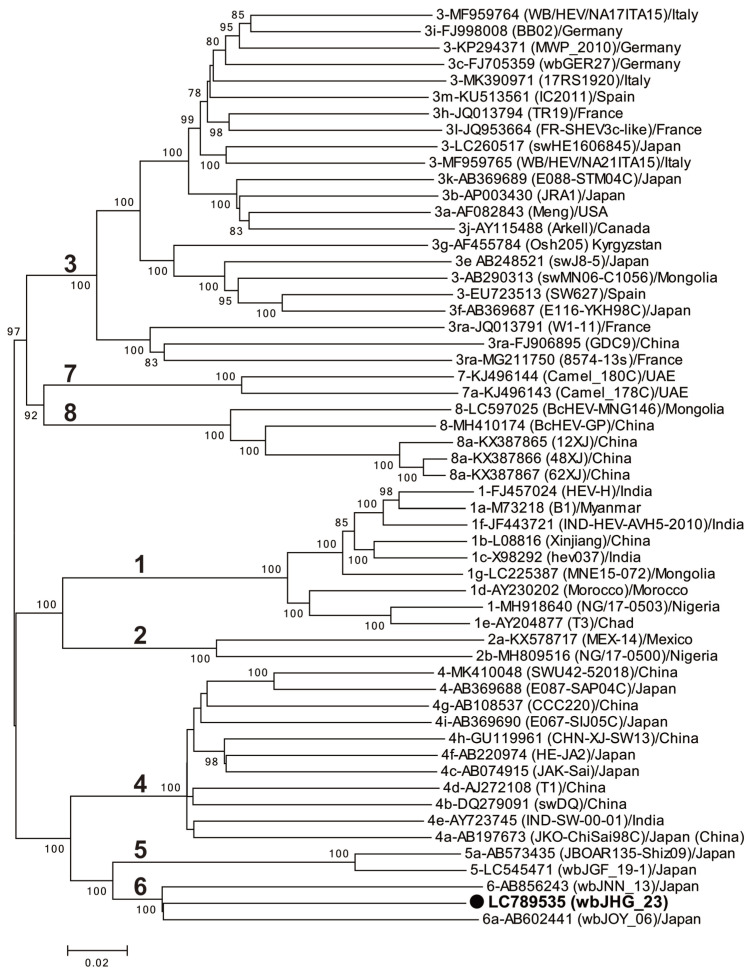
A maximum-likelihood phylogenetic tree of the entire genomic sequence of 55 reference HEV-1–8 strains proposed by Smith et al. [35], including additional HEV-5 and HEV-8 strains for which entire genomic sequences have been determined, as well as the HEV-6 strain (wbJHG_23) identified in our present study, which is denoted in bold typeface and marked with a closed circle for clarity. Each reported HEV strain is denoted with its genotype/subtype, DDBJ/EMBL/GenBank accession number, strain name in parentheses, and the name of the country where the HEV strain was isolated. The construction of the tree employed the maximum-likelihood method within the MEGA11 software program [34], optimizing tree topology and branch lengths. The values (>70%) on branches represent the percentage of 1000 full maximum-likelihood bootstrap replicates supporting the branch existence. A scale bar of 0.02 nucleotide substitutions per site is indicated.

**Figure 4 viruses-16-00842-f004:**
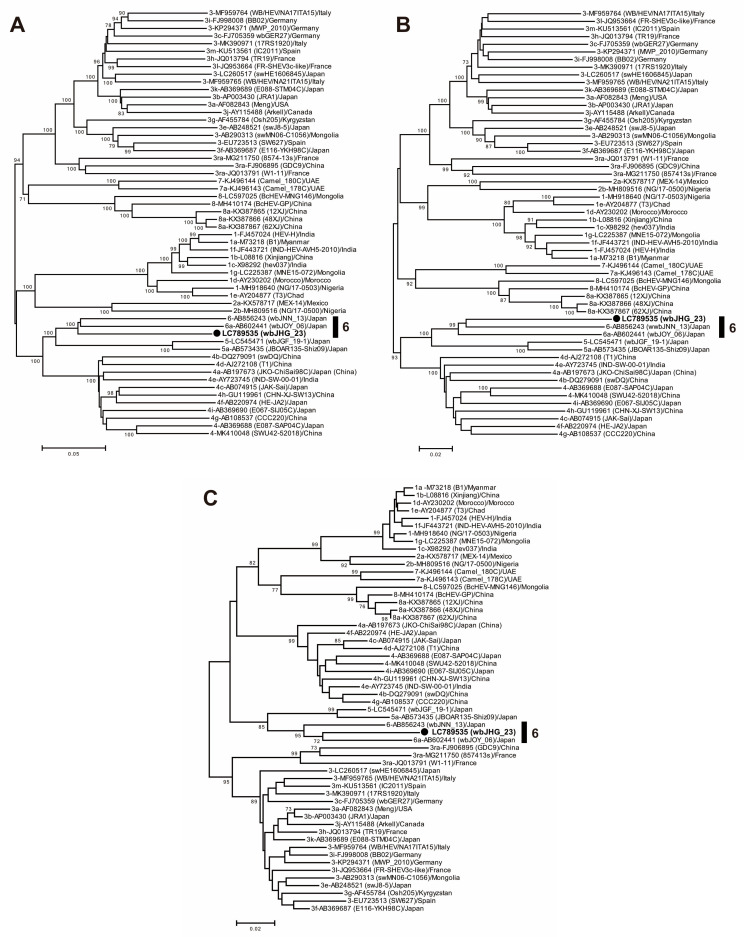
Maximum-likelihood phylogenetic trees of the ORF1 (**A**), ORF2 (**B**), and ORF3 (**C**) of 55 reference HEV-1–8 strains proposed by Smith et al. [35], including additional HEV-5 and HEV-8 strains for which entire genomic sequences have been determined, as well as the HEV-6 strain (wbJHG_23) obtained in the present study, which is highlighted in bold typeface and denoted by a closed circle for clarity. Strains are annotated with the genotype/subtype, DDBJ/EMBL/GenBank accession number, strain name in parentheses, and their country of isolation. The trees were constructed by employing the maximum-likelihood method within the MEGA11 software program [34], optimizing both tree topology and branch lengths. Numerical values (>70%) on tree branches represent the percentage of 1000 maximum-likelihood bootstrap replicates supporting the existence of each branch. The scale bar indicates 0.02 or 0.05 nucleotide substitutions per site.

**Figure 5 viruses-16-00842-f005:**
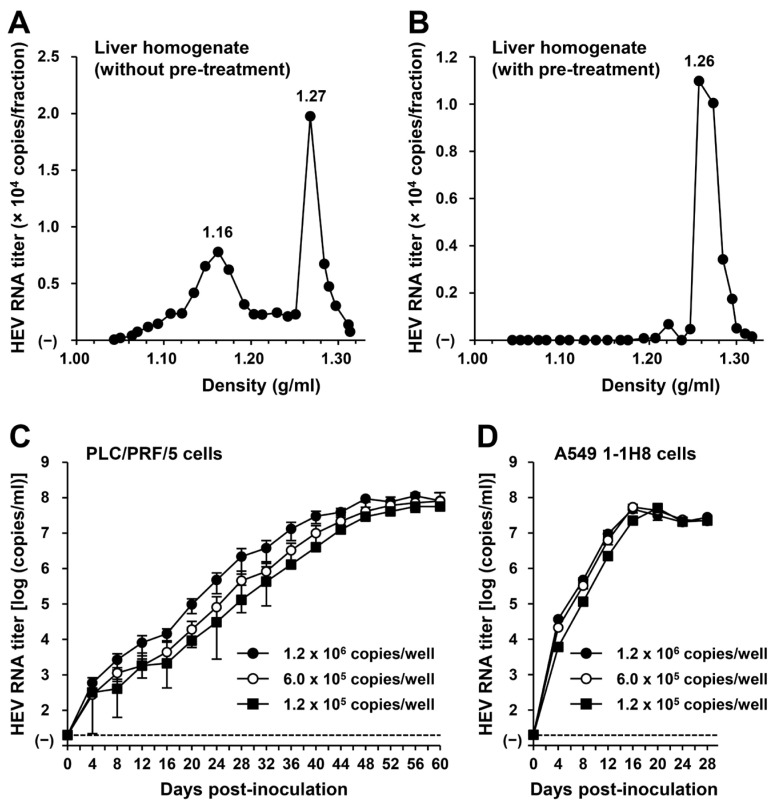
The ability of HEV-6 to infect human-derived cancer cell lines. (**A**,**B**) Sucrose density gradient fractionation of liver homogenate supernatants containing the HEV-6 (wbJHG_23 strain) without pre-treatment (**A**) and with pre-treatment by 0.2% sodium deoxycholate (DOC-Na) and 0.2% trypsin (**B**). (**C**,**D**) The quantification of HEV RNA in culture supernatants of PLC/PRF/5 (**C**) or A549 1-1H8 (**D**) cells inoculated with liver homogenate supernatants of wbJHG_23 at 1.2 × 10^6^ copies/well, 6.0 × 10^5^ copies/well, and 1.2 × 10^5^ copies/well. The virus growth was observed until peak HEV RNA titer in culture supernatants was achieved. The data are presented as the mean ± standard deviation (SD) for three wells each from a single experiment. The dotted horizontal line represents the limit of detection via real-time reverse transcription–polymerase chain reaction (RT-PCR) in the current study at 2.0 × 10^1^ copies/mL. (**E**) A Western blot analysis of the HEV ORF2 and ORF3 proteins in the culture supernatants and lysates of the inoculated PLC/PRF/5 cells at 60 days post-inoculation (dpi) with anti-HEV ORF2 monoclonal antibody (MAb) (TA8117) or anti-HEV ORF3 MAb (TA0536). (**F**) Indirect immunofluorescence staining of the HEV ORF2 and ORF3 proteins in the inoculated PLC/PRF/5 cells at 60 dpi. The cells were incubated with anti-HEV ORF2 MAb (H6225) or anti-HEV ORF3 MAb (TA0536) and then stained with Alexa-Fluor-488-conjugated anti-mouse IgG. Uninfected PLC/PRF/cells were used as negative controls. Nuclei were stained with 4′,6-diamidino-2-phenylindole dihydrochloride (DAPI). Bar, 50 µm. For Western blotting and immunofluorescence assays, results representative of two experiments are shown.

**Figure 6 viruses-16-00842-f006:**
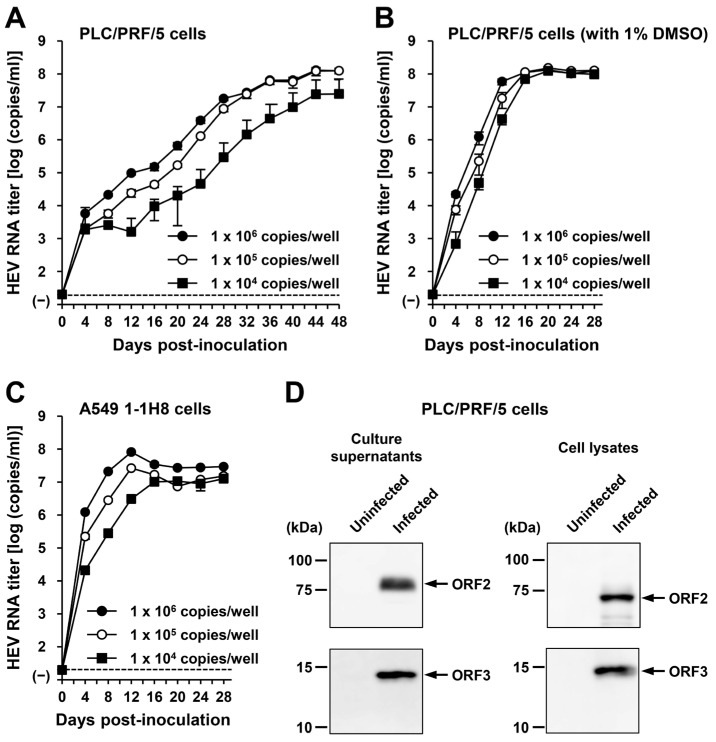
The ability of HEV-6 progenies to produce infectious virus in cultured cells. (**A**–**C**) The quantification of HEV RNA in culture supernatants of PLC/PRF/5 cells without DMSO supplementation (**A**) and with 1% DMSO supplementation (**B**) or A549 1-1H8 cells (**C**). The cells were inoculated with the culture supernatants of wbJHG_23-infected A549 1-1H8 cells at 1.0 × 10^6^ copies/well, 1.0 × 10^5^ copies/well, and 1.0 × 10^4^ copies/well. The virus growth was observed until peak HEV RNA titer in culture supernatants was achieved. The data are presented as the mean ± SD for three wells each from a single experiment. The dotted horizontal line represents the limit of detection via RT-PCR in the current study, set at 2.0 × 10^1^ copies/mL. (**D**) A Western blot analysis of the HEV ORF2 and ORF3 proteins in the culture supernatants and lysates of the inoculated PLC/PRF/5 cells at 28 dpi with anti-HEV ORF2 MAb (TA8117) or anti-HEV ORF3 MAb (TA0536). Results representative of two experiments are shown.

**Figure 7 viruses-16-00842-f007:**
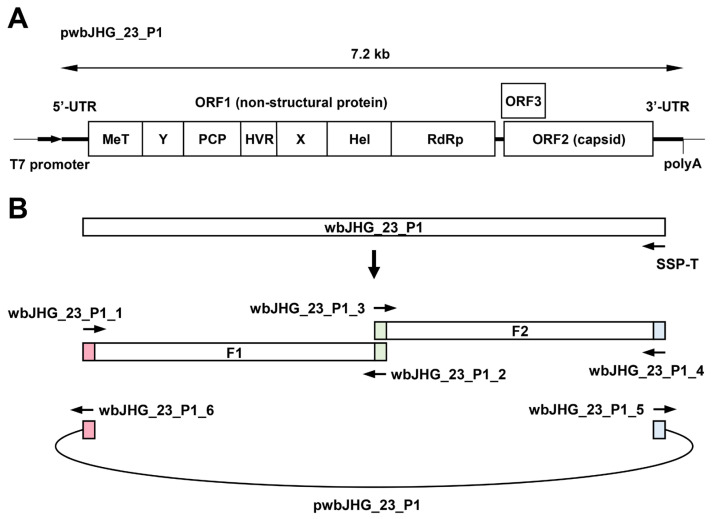
Construction of an infectious cDNA clone of HEV-6. (**A**) A schematic representation of the full-length genome of the wbJHG_23_P1 strain. (**B**) The strategy used to construct the full-length cDNA clone (pwbJHG_23_P1). Two fragments (F1 and F2) covering its whole genome were generated via RT-PCR and then cloned into the pUC19 vector in a stepwise manner using the In-Fusion cloning method. The 15 base pairs (bp) that overlap at their ends are highlighted with the same colors. UTR, untranslated region; MeT, methyltransferase; Y, Y domain; PCP, papain-like cysteine protease; HVR, hypervariable region; X, X or macro domain; Hel, helicase; RdRp, RNA-dependent RNA polymerase.

**Figure 8 viruses-16-00842-f008:**
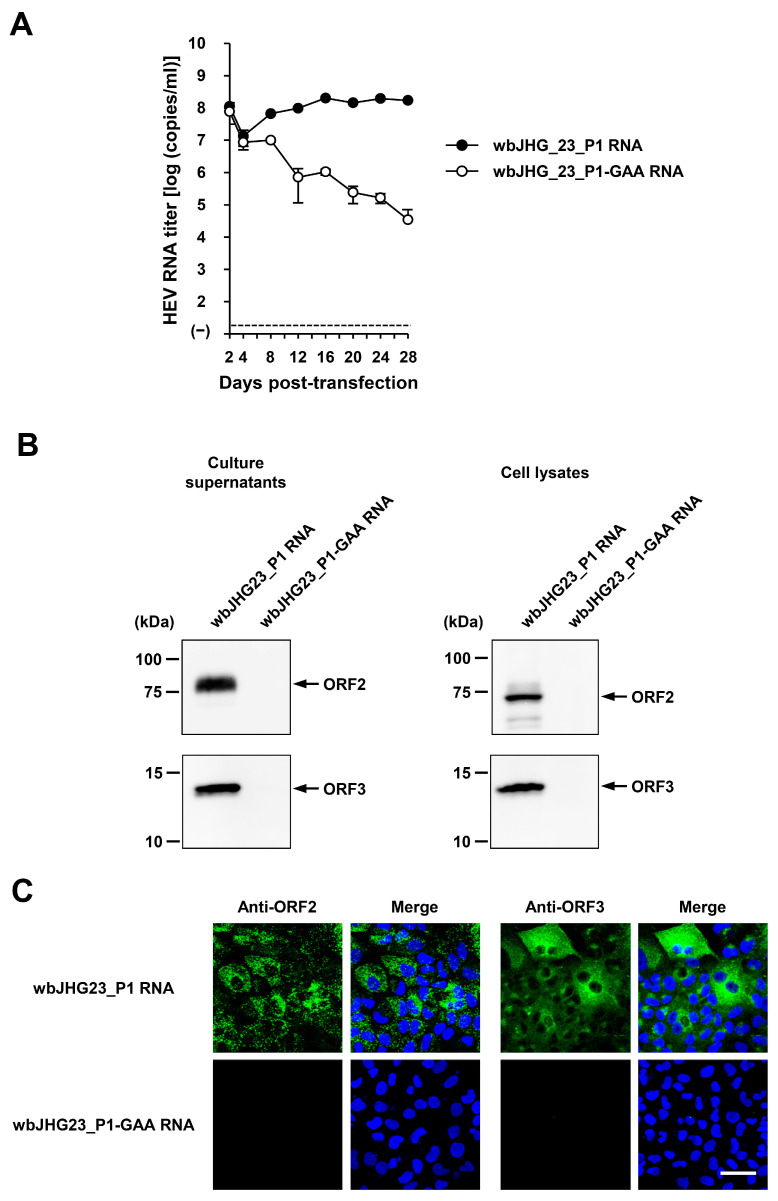
The replication ability of the cDNA clone of wbJHG_23_P1. (**A**) The quantification of HEV RNA in culture supernatants of PLC/PRF/5 cells transfected with RNA transcripts of pwbJHG_23_P1 or its replication-defective mutant (pwbJHG_23_P1-GAA), which served as a negative control. HEV growth was observed for 28 days. The data are presented as the mean ± SD for three wells each from a single experiment. The dotted horizontal line represents the limit of detection via real-time RT-PCR in the current study, set at 2.0 × 10^1^ copies/mL. (**B**) A Western blot analysis of the HEV ORF2 and ORF3 proteins in the culture supernatants and lysates of the transfected PLC/PRF/5 cells at 28 days post-transfection (dpt) with anti-HEV ORF2 MAb (TA8117) or anti-HEV ORF3 MAb (TA0536). (**C**) Indirect immunofluorescence staining of the HEV ORF2 and ORF3 proteins in the transfected PLC/PRF/5 cells from 28 dpt. The cells were incubated with anti-HEV ORF2 MAb (H6225) or anti-HEV ORF3 MAb (TA0536) and then stained with Alexa-Fluor-488-conjugated anti-mouse IgG. PLC/PRF/cells transfected with RNA transcripts of pwbJHG_23_P1-GAA were used as negative controls. Nuclei were stained with DAPI. Bar, 50 µm. For Western blotting and immunofluorescence assays, results representative of two experiments are shown.

**Figure 9 viruses-16-00842-f009:**
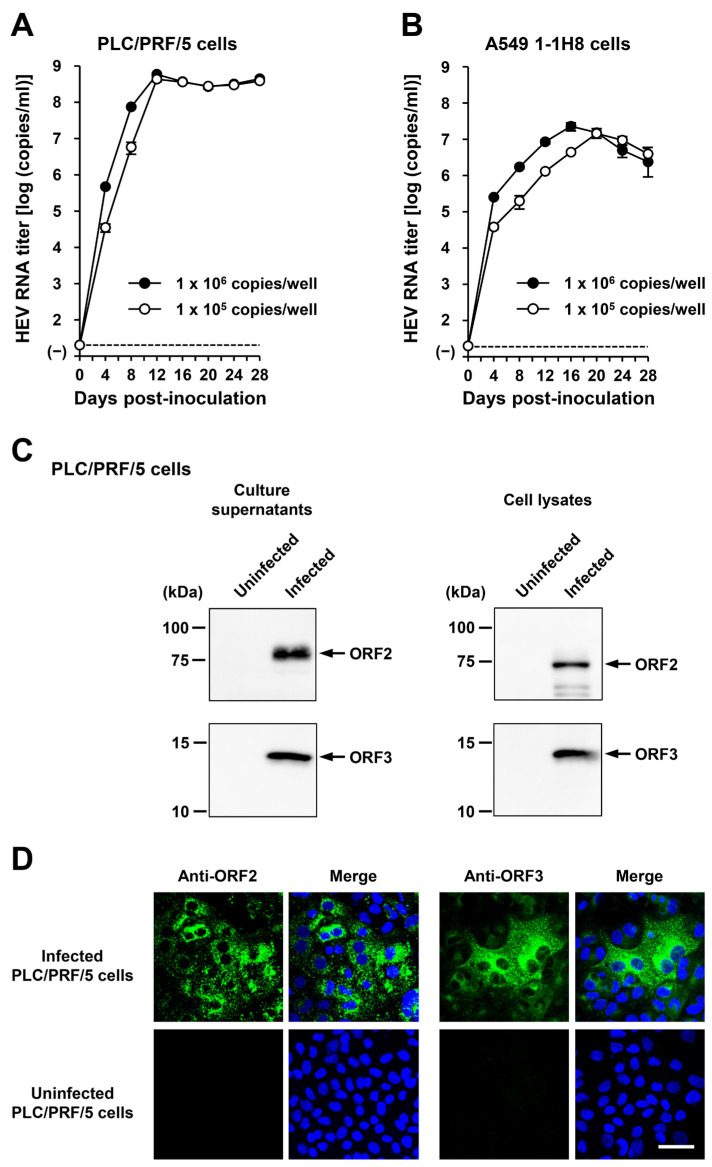
The infectivity of cDNA-derived wbJHG_23_P1 progenies to PLC/PRF/5 and A549 1-1H8 cells. (**A**,**B**) The quantification of HEV RNA in culture supernatants of PLC/PRF/5 (**A**) or A549 1-1H8 (**B**) cells inoculated with cDNA-derived wbJHG_23_P1 progeny viruses generated in culture supernatants at 1.0 × 10^6^ copies/well and 1.0 × 10^5^ copies/well. The virus growth was observed for 28 days. The data are presented as the mean ± SD for three wells each from a single experiment. The dotted horizontal line represents the limit of detection via RT-PCR in the current study, set at 2.0 × 10^1^ copies/mL. (**C**) A Western blot analysis of the HEV ORF2 and ORF3 proteins in the culture supernatants and lysates of the inoculated PLC/PRF/5 cells at 28 dpi with anti-HEV ORF2 MAb (TA8117) or anti-HEV ORF3 MAb (TA0536). (**D**) Indirect immunofluorescence staining of the HEV ORF2 and ORF3 proteins in the inoculated PLC/PRF/5 cells from 28 dpi. The cells were incubated with anti-HEV ORF2 MAb (H6225) or anti-HEV ORF3 MAb (TA0536) and then stained with Alexa-Fluor-488-conjugated anti-mouse IgG. Uninfected PLC/PRF/cells were used as negative controls. Nuclei were stained with DAPI. Bar, 50 µm. For Western blotting and immunofluorescence assays, results representative of two experiments are shown.

**Figure 10 viruses-16-00842-f010:**
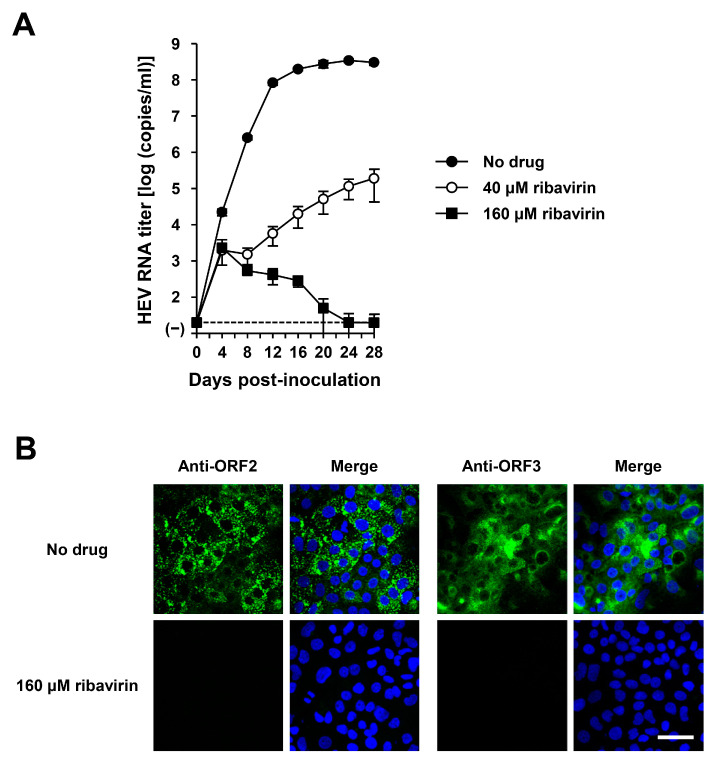
The sensitivity of HEV-6 to ribavirin in the cultured cells. (**A**) The quantification of HEV RNA in culture supernatants of PLC/PRF/5 cells inoculated with cDNA-derived wbJHG_23_P1 progeny viruses at 1.0 × 10^5^ copies/well in the presence of 40 µM or 160 µM ribavirin in DMSO (final concentration, 1%). HEV growth kinetics were observed for 28 days. The data are presented as the mean ± SD for four wells each from a single experiment. The dotted horizontal line represents the limit of detection via real-time RT-PCR used in the current study, set at 2.0 × 10^1^ copies/mL. (**B**) Indirect immunofluorescence staining of the HEV ORF2 and ORF3 proteins in the inoculated PLC/PRF/5 cells with or without 160 µM ribavirin treatment from 28 dpi. The cells were incubated with anti-HEV ORF2 MAb (H6225) or anti-HEV ORF3 MAb (TA0536) and then stained with Alexa-Fluor-488-conjugated anti-mouse IgG. Infected PLC/PRF/cells with no ribavirin treatment were used as negative controls. Nuclei were stained with DAPI. Bar, 50 µm. Results representative of two experiments are shown.

**Table 1 viruses-16-00842-t001:** The primers used to construct pwbJHG_23_P1 and pwbJHG_23_P1-GAA cDNA clones.

Name	Polarity	Sequence (5′ to 3′)	Note
SSP-T		AAGGATCCGTCGACATCGATAATACGTTTTTTTTTTTTTTT	cDNA synthesis
wbJHG_23_P1_1	+	GCTTAATACGACTCACTATAGCAGGCCACATGTGGCCGACGCC	T7 promoter (underlined) and positive-strand sequence (nt 1–23 ^a^)
wbJHG_23_P1_2	–	GCATATGAAGCAGGAGCAGGTGC	Negative-strand sequence (nt 3182–3204)
wbJHG_23_P1_3	+	CCTGCTCCTGCTTCATATGCAGC	Positive-strand sequence (nt 3185–3207)
wbJHG_23_P1_4	–	GCCCCAAGGGGTTATGCTAGTTTTTTTTTTTTTTTTTTTTTTTTTTTTTTTCCGGGAGCGCGGAACGCAGAAAAAGG	pUC19 vector, poly(A), and negative-strand sequence (nt 7219–7244)
wbJHG_23_P1_5	+	CTAGCATAACCCCTTGGGGCCTC	pUC19 vector
wbJHG_23_P1_6	–	TATAGTGAGTCGTATTAAGCTTGGCG	T7 promoter (underlined) and pUC19 vector
wbJHG_23_P1_*Avr*II-F	+	TCCGAGAGTCCCTAGGCCGG	Positive-strand sequence (nt 4054–4073), *Avr*II site (underlined)
wbJHG_23_P1-GAA-R	–	ACGGAGGCGGCACCCTTGAAGGCGGCGACC	Negative-strand sequence (nt 4706–4735) and mutated nucleotide (underlined)
wbJHG_23_P1-GAA-F	+	GGGTGCCGCCTCCGTTCTGCTCTGCAGTG	Positive-strand sequence (nt 4721–4749) and mutated nucleotide (underlined)
wbJHG_23_P1_*Stu*I-R	–	GGGCATAGTTAGAGGCCTCAG	Negative-strand sequence (nt 5724–5744) and *Stu*I site (underlined)

^a^ The nucleotide positions are numbered in accordance with the wbJHG_23 strain obtained in the present study.

**Table 2 viruses-16-00842-t002:** A comparison of the identity (%) within the partial ORF2 sequences (412 nucleotides) of the wbJHG_23 isolate obtained in the present study with the entire sequences of previously reported reference HEV isolates.

HEV Isolate	No. of Isolates Compared ^a^	Identity (%)
HEV-1	9	77.7−81.8 (79.3 ± 1.4)
HEV-2	2	77.2−77.7 (77.4 ± 0.3)
HEV-3	22	77.4−81.6 (79.3 ± 1.2)
HEV-4	11	77.9−80.6 (79.7 ± 0.9)
HEV-5	2	79.6−81.6 (80.6 ± 1.4)
HEV-6	2	82.5−83.0 (82.8 ± 0.3)
HEV-7	2	76.7−77.4 (77.1 ± 0.5)
HEV-8	5	76.9−79.9 (78.9 ± 1.2)

^a^ Reference HEV isolates for HEV genotype/subtype proposed by Smith et al. [35], including additional HEV-5 and HEV-8 strains whose entire genomic sequences have been determined. ORF2, open reading frame; HEV, hepatitis E virus.

**Table 3 viruses-16-00842-t003:** A comparison of the identity (%) over the entire genome and the ORF1, ORF2, or ORF3 sequences of the wbJHG_23 isolate obtained in the present study with the entire sequences of previously reported reference HEV isolates.

HEV Isolate ^a^	Number of Isolates Compared ^b^	Identity (%)			
		Entire Genome	ORF1	ORF2	ORF3
HEV-1	9	73.9−74.9 (74.3 ± 0.3)	72.1−73.1 (72.4 ± 0.3)	77.7−79.1 (78.3 ± 0.4)	83.5−85.1 (84.3 ± 0.5)
HEV-2	2	74.0−74.1 (74.1 ± 0.0)	72.6−73.0 (72.8 ± 0.3)	76.8−77.2 (77.0 ± 0.3)	82.9−83.8 (83.4 ± 0.6)
HEV-3	22	73.5−75.0 (74.4 ± 0.3)	71.9−73.7 (72.9 ± 0.4)	77.2−79.5 (78.1 ± 0.6)	80.2−85.4 (83.4 ± 1.1)
HEV-4	11	76.9−77.4 (77.2 ± 0.1)	75.4−76.5 (76.0 ± 0.3)	79.3−81.1 (80.2 ± 0.5)	85.7−87.8 (87.0 ± 0.6)
HEV-5	2	77.5−78.1 (77.8 ± 0.4)	76.0−76.9 (76.4 ± 0.6)	80.8−81.3 (81.0 ± 0.4)	86.0−87.5 (86.7 ± 1.1)
HEV-6	2	80.3−80.9 (80.6 ± 0.4)	79.1−79.4 (79.2 ± 0.2)	83.7−83.9 (83.8 ± 0.1)	89.6−90.6 (90.1 ± 0.7)
HEV-7	2	74.4−74.5 (74.5 ± 0.0)	73.0−73.2 (73.1 ± 0.2)	77.3−78.4 (77.4 ± 0.2)	80.5−81.1 (80.8 ± 0.4)
HEV-8	5	73.3−73.8 (73.5 ± 0.2)	71.3−71.9 (71.6 ± 0.2)	77.2−78.2 (77.7 ± 0.4)	80.8−83.2 (82.4 ± 1.1)

^a^ See Figure 3 for accession numbers. ^b^ Reference HEV isolates for HEV genotype/subtype proposed by Smith et al. [35], including additional HEV-5 and HEV-8 strains whose entire genomic sequences have been determined. ORF, open reading frame; HEV, hepatitis E virus.

**Table 4 viruses-16-00842-t004:** A comparison of the sequence of liver homogenate-derived wild-type (wt) wbJHG_23 strain and its cDNA clone of a cell-culture-produced variant (wbJHG_23_P1) over the entire genome.

Nucleotide Position	Region (Domain)	Nucleotide		Amino Acid	
		wt	pwbJHG_23_P1	Position	Substitution
587	ORF1 (MeT)	T	C	-	-
1659	ORF1 (PCP)	G	A	546	Gly → Ser
2879	ORF1 (X)	T	A	-	-
3377	ORF1 (Hel)	G	A	-	-
3848	ORF1 (RdRp)	G	A	-	-
4229	ORF1 (RdRp)	C	T	-	-
4793	ORF1 (RdRp)	C	T	-	-
5461	ORF3	C	T	92	Thr → Ile
6241	ORF2	T	C	-	-

ORF, open reading frame; MeT, methyltransferase; PCP, papain-like cysteine protease; X, X or macro domain; Hel, helicase; RdRp, RNA-dependent RNA polymerase.

## Data Availability

All data generated or analyzed during this study are either incorporated within this published article or accessible upon reasonable request to the corresponding author. The nucleotide sequence data reported in this study have been assigned the DDBJ/EMBL/GenBank accession numbers LC789535, LC815003, and LC815004.

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
