# Peer review of "The Full-Genome Analysis and Generation of an Infectious cDNA Clone of a Genotype 6 Hepatitis E Virus Variant Obtained from a Japanese Wild Boar: In Vitro Cultivation in Human Cell Lines"

_viruses, 2024, doi:10.3390/v16060842_

Round 1
Reviewer 1 Report
Comments and Suggestions for Authors
In the present manuscript, the authors have characterised a strain of genotype 6 hepatitis E virus (HEV-6) originating from a wild boar in Japan (wbJHG_23). They have successfully infected human cells with wbJHG_23 and constructed a cDNA infectious clone that represents an interesting tool for further studies. This paper is of interest for the field as it can be difficult to develop cell culture systems to study HEV and the zoonotic potential of HEV-6 remains to be confirmed. The paper is clear and well-written. Here are few minor comments for consideration:
- Line 60-61: “Other animals reported to cross the species barrier to cause human infection include wild boars, deer, rabbits, camels, and rats “ : this sentence is not clear and should be rewritten.
- Figure 4: The size of this figure should be increased for clarity
- Table 4 should be inserted earlier within the manuscript (before figure 7)
- Line 570-571 and line 582: expression of the proteins does not mean that the HEV-6 cDNA clone can produce infectious particles but only that it is replicating. It is only confirmed later within the manuscript when cells are infected with cDNA-derived progenies (figure 9)
- Line 602 and 625 : where are the progeny viruses coming from ? Supernatant or lysates of transfected cells ?
- Line 711: reference is missing for “ similar to observations in other wild boar-derived HEV genotypes”
- Description of the results should be avoided in the discussion section (for example line 701-709).
- Line 745-751: Have you tried to re-introduced the 2 missing nt in the 5’ UTR region to really determine whether this deletion impact replication and infectivity?
Comments on the Quality of English LanguageEnglish is of good quality.
Author Response
Responses to Reviewer 1
Comments and Suggestions for Authors
In the present manuscript, the authors have characterised a strain of genotype 6 hepatitis E virus (HEV-6) originating from a wild boar in Japan (wbJHG_23). They have successfully infected human cells with wbJHG_23 and constructed a cDNA infectious clone that represents an interesting tool for further studies. This paper is of interest for the field as it can be difficult to develop cell culture systems to study HEV and the zoonotic potential of HEV-6 remains to be confirmed. The paper is clear and well-written. Here are few minor comments for consideration:
Response: Thank you very much for your favorable comments and suggestions. Point-to-point responses to the comments are described below.
- Line 60-61: “Other animals reported to cross the species barrier to cause human infection include wild boars, deer, rabbits, camels, and rats “ : this sentence is not clear and should be rewritten.
Response: Thank you for your comment. The part has been revised to “Most animal-derived HEV strains have been isolated from pigs. Besides pig-derived HEV strains, which can cross the species barrier and infect humans, HEV strains from other animals, including wild boars, deer, rabbits, camels, and rats, are also capable of causing zoonotic infections” (Lines 60–63).
- Figure 4: The size of this figure should be increased for clarity
Response: Thank you for your suggestion. The figure has been revised accordingly.
- Table 4 should be inserted earlier within the manuscript (before figure 7)
Response: Thank you for your suggestion. Table 4 has been inserted before Figure 7 within the manuscript.
- Line 570-571 and line 582: expression of the proteins does not mean that the HEV-6 cDNA clone can produce infectious particles but only that it is replicating. It is only confirmed later within the manuscript when cells are infected with cDNA-derived progenies (figure 9)
Response: Thank you for your thoughtful comment. The sentences have been revised accordingly: “Taken together, these results demonstrate replication ability of the HEV-6 cDNA clone” (Lines 537–538); “Replication ability of the cDNA clone of wbJHG_23_P1” (Line 572); and “A Western blotting analysis and immunofluorescence assays confirmed the intracellular and extracellular expression of ORF2 and ORF3 proteins (Figure 8B, C), validating the replication ability of the wbJHG_23_P1 cDNA clone” (Lines 693-694).
- Line 602 and 625 : where are the progeny viruses coming from ? Supernatant or lysates of transfected cells ?
Response: Thank you for your comment. The sentences have been revised accordingly: “To examine whether or not the cDNA-derived wbJHG_23_P1 progenies generated in the culture supernatants were infectious, they were inoculated into PLC/PRF/5 (Figure 9A) and A549 1-1H8 (Figure 9B) cells at viral inoculum titers of 1.0 × 106 and 1.0 × 105 copies/well” (Lines 546–549); “(A, B) Quantification of HEV RNA in culture supernatants of PLC/PRF/5 (A) or A549 1-1H8 (B) cells inoculated with cDNA-derived wbJHG_23_P1 progeny viruses generated in culture supernatants at 1.0 × 106 copies/well and 1.0 × 105 copies/well” (Lines 589–590); and “Furthermore, the progenies in culture supernatants derived from wbJHG_23_P1 cDNA clone were infectious (Figure 9)” (Lines 694–695).
- Line 711: reference is missing for “ similar to observations in other wild boar-derived HEV genotypes”
Response: Thank you for your comment. The references have been added to the sentence accordingly: “The ability of HEV-6 to infect human-derived cancer cell lines underscores its potential to cause human infections, similar to observations in other wild boar-derived HEV genotypes [26, 48-55]” (Lines 663–665).
- Description of the results should be avoided in the discussion section (for example line 701-709).
Response: Thank you for your kind suggestion. The Discussion section has been revised accordingly.
- Line 745-751: Have you tried to re-introduced the 2 missing nt in the 5’ UTR region to really determine whether this deletion impact replication and infectivity?
Response: Thank you for your comment. We have not tried to re-introduce the missing 2 nt in the 5’-UTR, since in the determination of 5’-UTR sequence (by cloning), all generated clones harbor the same 2 nt deletion. In addition, the wild-type virus and the cDNA clone-derived virus progenies can replicate efficiently in cultured cells. Therefore, the 2 nt deletion might be a characteristic of this wbJHG_23 strain. The impact of this deletion in the 5’-UTR on HEV replication and infectivity would be an interesting topic for future investigation. We revised the last part of this paragraph accordingly in Lines 704–705.

Reviewer 2 Report
Comments and Suggestions for Authors
Title: A Full-genome Analysis and Generation of Infectious cDNA 2 Clone of a Genotype 6 Hepatitis E Virus Variant Obtained from 3 a Japanese Wild Boar: In Vitro Cultivation in Human Cell Lines
HEV is an important emerging but understudied zoonotic virus. One of its’ significant features is broad range of animal reservoirs and huge genetic diversity forming many different genotypic virus stains. This study found and isolated a novel genotype 6 HEV virus strain from wild boar in Japan. The authors did comprehensive works to investigate the features of the new virus stain, including genome analysis, cell culture system establishment, and infectious clone construction. Their work made a significant contribution to the field of HEV research. The manuscript is well organized and written. No major concern from this reviewer, only one error needed to be addressed.
Lines 448-449, 453, the number should be x106 , x105, x105. Correct it in the text and in the figure legend (fig5).
Overall redundant descriptions for M&M and Results. Suggest a more concise writing would make the manuscript more readable.
Author Response
Responses to Reviewer 2
Comments and Suggestions for Authors
Title: A Full-genome Analysis and Generation of Infectious cDNA 2 Clone of a Genotype 6 Hepatitis E Virus Variant Obtained from 3 a Japanese Wild Boar: In Vitro Cultivation in Human Cell Lines
HEV is an important emerging but understudied zoonotic virus. One of its’ significant features is broad range of animal reservoirs and huge genetic diversity forming many different genotypic virus stains. This study found and isolated a novel genotype 6 HEV virus strain from wild boar in Japan. The authors did comprehensive works to investigate the features of the new virus stain, including genome analysis, cell culture system establishment, and infectious clone construction. Their work made a significant contribution to the field of HEV research. The manuscript is well organized and written. No major concern from this reviewer, only one error needed to be addressed.
Response: Thank you very much for your thoughtful comments and suggestions. Point-to-point responses are described below.
Lines 448-449, 453, the number should be x106 , x105, x105. Correct it in the text and in the figure legend (fig5).
Response: Thank you for pointing out. The sentences have been corrected: “… for inoculum titers of 1.2 × 106, 6.0 × 105, and 1.2 × 105 copies/well, respectively (Figure 5C)” Lines 425–426); “…5.0 × 107 copies/ml at 20 dpi for inoculum titers of 1.2 × 106, 6.0 × 105, and 1.2 × 105 copies/well, respectively” (Lines 430–431); and “…(D) cells inoculated with liver homogenate supernatants of wbJHG_23 at 1.2 × 106 copies/well, 6.0 × 105 copies/well, and 1.2 × 105 copies/well” (Lines 452–453).
Overall redundant descriptions for M&M and Results. Suggest a more concise writing would make the manuscript more readable.
Response: Thank you for your kind suggestion to improve our manuscript. The Materials and Methods as well as Results Sections have been revised accordingly.